

**Uncertainty quantification in application of linear lumped**
**rainfall-runoff models**
Ching-Min Chang and Hund-Der Yeh
Institute of Environmental Engineering, National Chiao Tung University,
Hsinchu, Taiwan
*Correspondence to:* Hund-Der Yeh (hdyeh@mail.nctu.edu.tw)
**Abstract.**    This study proposes a stochastic framework for a linear lumped
rainfall-runoff problem at the catchment scale. An autoregressive (AR) model is adopted
to account for the temporal variability of the rainfall process. For a stochastic description,
solutions of the surface flow problem are derived in terms of first two statistical moments
of the runoff discharge through the nonstationary Fourier-Stieltjes representation
approach. The closed-form expression for the variance of runoff discharge allows to
assessing the impacts of rainfall and storage parameters, respectively, on the discharge
variability. It is found that the temporal variability of the runoff discharge induced by a
random rainfall process persists longer for smaller values of the storage or rainfall
parameters.

**1   Introduction**

Rainfall-runoff models simulate the processes of converting rainfall to runoff. They
are used for a variety of applications in hydrology (e.g., Beven, 2012; Falahi et al., 2012),



for example, to predict the peak flow used in drainage design purposes, to estimate flows
of ungauged catchments, to assess the effects of climate changes. The quantitation of
rainfall-runoff processes is essential for providing a basis of water resources management
and planning in river basins.

Rainstorm is the major input into the generation of surface runoff and the production

of runoff is, therefore, dependent on the characteristics of rainfall events. Rainfall
processes are generally recognized as being affected by complex natural events. The
details of the processes cannot be described precisely. Moreover, to carry out
rainfall-runoff calculations detailed information about landscape properties and
hydrologic states must be known in the whole catchment. In general, such information is
not available due to the heterogeneity in associated parameters. Therefore, there is a great
deal of uncertainty about the runoff prediction using a deterministic model. As such, the
analysis of rainfall-runoff processes is often taken by means of a stochastic framework
(e.g., Córdova and Rodríguez-Iturbe, 1985; Goel et al., 2000; Lee et al., 2001; Moore,
2007; Bartlett et al., 2016).

Much of stochastic research in rainfall-runoff modellings focused on development of

the probability distribution of state variables (such as rainfall and flow discharge). In
most cases, due to a complex non-linear behavior in general, the analytical solution for
the probability distribution function does not exist. Alternatively, to take the advantage of
closed-form expressions, the purpose of this study is to derive analytical solutions,
namely the first two moments of runoff discharge, for a linear lumped rainfall-runoff
problem. The first moment (ensemble mean) is used as an unbiased estimate of a system
state, and the second moment (ensemble variance) is used as a measure of uncertainty by



applying the mean model. Those expressions will be obtained using the nonstationary
Fourier-Stieltjes representation approach along with the assumption of an AR rainfall
model (e.g., Foufoula-Georgiou and Lettenmaier, 1987; Thyregod et al., 1999; Srikanthan,
and McMahon, 2001; Rebora et al. 2006; Hannachi, 2014).

**2   Mathematical Statement of the Problem**

The physical-based equation in modeling the rainfall-runoff process is the equation of
conservation of mass. If the control volume is extended to the scale of a catchment, the
continuity equation for the free surface flow then takes on the lumped form of the
storage equation as (e.g., Brutsaert, 2005; Beven, 2012)
$$\frac{dS}{dt} = R_t - E_t - Q \tag{1}$$
where $S$ is catchment storage, $R_t$ and $E_t$ denote the rainfall and evapotranspiration at time
$t$, respectively, and $Q$ is the discharge from the catchment. The lumped model attempts to
relate the forcing (rainfall input) to the model output (runoff) without considering the
spatial variability. Therefore, all variables and parameters in Eq. (1) represent spatial
averages over the entire catchment area, and, as such, only their temporal variability is
retained. That is, in a lumped system model, the flow is evaluated as a function of time
alone at a particular location in large catchments.
Since there are two unknowns, namely $Q$ and $S$, for only one equation, further
knowledge of the relation of $Q$ to $S$ is needed in order to solve Eq. (1). In most practical
applications, $S$ in Eq. (1) is specified as an arbitrary function of $Q$. As such, the changes
in $S$ with time may be expressed as





$$\frac{dS}{dt} = \frac{dS}{dQ}\frac{dQ}{dt}$$    (2)
Given Eqs. (1) and (2), it follows that
$$\frac{dQ}{dt} + \frac{Q}{dS/dQ} = \frac{R_t - E_t}{dS/dQ}$$    (3)
This study will concentrate only on the case of $S$ being a linear function of $Q$ (e.g.,
Kaseke and Thompson, 1997; Botter et al., 2007; Suweis et al., 2010, Guinot et al.,

2015):

$$S = KQ$$    (4)
where the constant $K$ is termed as the storage parameter. Consequently, Eq. (1) can be
cast in the form
$$\frac{dQ}{dt} + \frac{Q}{K} = \frac{R_t - E_t}{K}$$    (5)
It is assumed in the following analysis that $R_t$ is a temporal stochastic process (random
field). We also assume that evapotranspiration has a negligible effect on $Q$ as compared
to that of rainfall ($R_t \gg E_t$). Since the temporal random heterogeneity of $R_t$ appears as a
forcing term which generates the random variations in $Q$, the differential Eq. (5) is then
viewed as a stochastic differential equation. The probabilistic structure of random $Q$ is
determined by its temporal statistical moments. In the present study, we are interested
mainly in developing the first two moments of $Q$. The mean (unbiased estimate of)
runoff discharge may also be interpreted as the solution predicted by the deterministic
model. The second moment (variance) of catchment discharge derived below can then be
used to characterize the uncertainty in applying the deterministic (or mean) model. The
variance can be viewed as an index of large-scale discharge variability as well.



Due to its linearity, Eq. (5) may be split into two sub-equations: a mean equation
governing the temporal behavior of mean catchment discharge,
$$\frac{d\overline{Q}}{dt} + \frac{\overline{Q}}{K} = \frac{\overline{R}}{K}$$    (6a)
and an equation for the perturbations describing the discharge perturbation produced as a
result of the input rainfall perturbation,
$$\frac{dq}{dt} + \frac{q}{K} = \frac{r}{K}$$    (6b)
In Eq. (6), $\overline{Q}$ and $\overline{R}$ indicate the means of $Q$ and $R_t$, respectively, and $q$ ($= Q\text{-}\overline{Q}$) and $r$
($= R_t\text{-}\overline{R}$) are zero-mean perturbations.
Spectral representation theorem provides a very useful way of evaluating the
variance of perturbations. To carry out the calculation, the perturbed-form Eq. (6b) must
be solved in Fourier space. Since $r(t)$ in Eq. (6b) is a noise force contributing to the
variations in $q$, the solution of Eq. (6b) requires knowledge of the temporal distribution of
rainfall field. The section that follows attempts to develop the spectrum of $r(t)$ which will
be achieved by solving an AR model for temporal rainfall processes through the
nonstationary spectral approach.

**3    Spectral Solution for the Rainfall field**

The AR model specifies linear dependence of the output variable partly on its own
previous values and partly on the random disturbance (or white noise) (e.g., Priestley,
1981; Vanmarcke, 1983). In other words, the AR model uses a linear equation with
constant coefficients to define the relation between an output process and an input white





noise process.
Throughout this study, it is assumed that the temporal distribution of rainfall field can
be described by the AR model proposed by Vanmarcke (1983). Following Vanmarcke
(1983), the random rainfall perturbation field $r(t)$ without directional preference may be
expressed in the form
$r(t) = a[r(t-1) + r(t+1)] + \xi(t)$ (7a)
where $a$ is a constant parameter and $\xi$ is a stationary purely random (white noise) process.
Subtracting $2ar(t)$ from both sides and rearranging terms yields (*Vanmarcke*,1983)
$a[r(t-1) - 2r(t) + r(t+1)] - (1-2a)r(t) = \xi(t)$ (7b)
In continuous time, the natural analogue of the linear Eq. (7b) is a linear differential
equation, of the form
$\dfrac{d^2 r}{dt^2} - \alpha^2 r = \xi(t)$ (8)
where $\alpha^2 = (1-2a)/a$. In addition, the initial conditions are specified as
$r(0) = 0$ (9a)
$\dfrac{d}{dt} r(0) = 0$ (9b)
Eq. (8) along with Eq. (9) permits one to determine the spectrum of $r(t)$.
Whenever the random field is stationary, there always exists an unique
representation of the process in terms of a Fourier-Stieltjes integral as (e.g., Lumley and
Panofsky, 1964)
$\xi(t) = \displaystyle\int_{-\infty}^{\infty} e^{i\omega t} dZ_{\xi}(\omega)$ (10)




where $Z_\xi(\omega)$ is an orthogonal process (i.e., the random amplitudes $dZ_\xi$ are uncorrelated)
and $\omega$ denotes the frequency. Without the restriction that the $r(t)$ process must be
stationary, the perturbed quantities $r(t)$ may be presented as (Priestley, 1965)
$$r(t) = \int_{-\infty}^{\infty} \Lambda_{r\xi}(t;\omega)\, e^{i\omega t}\, dZ_\xi(\omega)$$   (11)
In Eq. (11), $\Lambda_{r\xi}(\text{-})$ is referred to as the modulating function by Priestley (1965).
Introducing Eqs. (10) and (11) into Eqs. (8) and (9), respectively, produces
$$\frac{d^2 \Lambda_{r\xi}}{dt^2} + i2\omega \frac{d\,\Lambda_{r\xi}}{dt} - (\omega^2 + \alpha^2)\,\Lambda_{r\xi} = 1$$   (12)
with
$$\Lambda_{r\xi}(0;\omega) = 0$$   (13a)
$$\frac{d\Lambda_{r\xi}(0;\omega)}{dt} = 0$$   (13b)
The system of equations admits the solution as follows:
$$\Lambda_{r\xi}(t;\omega) = \frac{1}{\alpha^2 + \omega^2}\left[-1 + \frac{\alpha + i\omega}{2\alpha} e^{\eta - i\tau} + \frac{\alpha - i\omega}{2\alpha} e^{-\eta - i\tau}\right]$$   (14)
where $\eta = \alpha t$ and $\tau = \omega t$. Using Eq. (14), Eq. (11) implies
$$r(t) = \int_{-\infty}^{\infty} \frac{1}{\alpha^2 + \omega^2}[-e^{i\tau} + \frac{\alpha + i\omega}{2\alpha} e^{\eta} + \frac{\alpha - i\omega}{2\alpha} e^{-\eta}]dZ_\xi(\omega)$$   (15)

It follows from using the representation theorem for $r(t)$ that the variance of $r(t)$, $\sigma_r^2$,

admits a representation of the form



$$\sigma_r^2(t) = E[r(t)\, r^*(t)] = \int_{-\infty}^{\infty} \Lambda_{r\xi}(t;\omega)\, \Lambda_{r\xi}^*(t;\omega)\, S_{\xi\xi}(\omega)\, d\omega = \int_{-\infty}^{\infty} S_{rr}(\omega)\, d\omega \qquad (16)$$
where $E[\cdot]$ indicates the ensemble average of the quantity, * denotes the complex
conjugate, $S_{\xi\xi}(\omega)$ is the spectrum of $\xi(t)$, and $S_{rr}(t;\omega)$ is the evolutionary spectrum of $r(t)$,
quantified corresponding to Eqs. (14) and (16) as
$$S_{rr}(t;\omega) = \frac{1}{\omega^4(1+\gamma^2)^2}\left[1 - 2\cos(\tau)\cosh(\eta) - \frac{2}{\gamma}\sin(\tau)\sinh(\eta) + \frac{1+\gamma^2}{2\gamma^2}\cosh(2\eta) + \frac{\gamma^2-1}{2\gamma^2}\right]S_{\xi\xi}(\omega) \qquad (17)$$
In Eq. (17), $\gamma = \alpha/\omega$. The evolutionary spectrum referred by Priestley (1965) has the same
physical interpretation as the spectrum of a stationary process that it describes the energy
of a signal distributed with frequency. The latter is determined by the behavior of the
process over all time, while the former represents specifically the spectral content of the
process in the neighborhood of the time instant $t$.

As defined above, $\xi(t)$ represents a white noise process which consists of a sequence

of uncorrelated random variables. The corresponding spectrum for such a process is
$$S_{\xi\xi}(\omega) = I_\xi \qquad (18)$$
$I_\xi$ in Eq. (18) is constant for all frequency. The variance of the rainfall field resulting
from Eqs. (16)-(18) is now given by
$$\sigma_r^2(t) = \frac{\pi}{2\alpha^3}\Gamma_t I_\xi \qquad (19)$$
where $\Gamma_t = \sinh(2\eta)-2\eta$.

It follows from Eqs. (17)-(19) that for a given $\sigma_r^2$, the evolutionary spectrum of the

rainfall response to white noise input can be rewritten as





$$S_{rr}(t;\omega) = \frac{2}{\pi} \frac{\gamma^3}{\omega(1+\gamma^2)^2} \Psi_t \sigma_r^2$$   (20)
with
$$\Psi_t = \frac{1}{\Gamma_t}\left[1 - 2\cos(\tau)\cosh(\eta) - \frac{2}{\gamma}\sin(\tau)\sinh(\eta) + \frac{1+\gamma^2}{2\gamma^2}\cosh(2\eta) + \frac{\gamma^2-1}{2\gamma^2}\right]$$   (21)
The dependence of $S_{rr}(t;\omega)$ in Eq. (20) on rainfall parameter $\alpha$ is depicted in Fig. 1 at
different times. The reduction of the temporal rainfall spectrum with $\alpha$ is clearly
observed in the figure. This reflects that a larger $\alpha$ produces shorter persistence of
rainfall perturbations, which, in turn, leads to less deviations of the rainfall perturbation
from the mean rainfall profile and, consequently, less variability of the rainfall process. It
can be shown that the variance of rainfall in Eq. (19) will decrease with a large $\alpha$.

The results presented in this section will be employed in the derivation of solutions

for the flow discharge problem in terms of its moments.

**4   Moments of discharge**

We consider the case where initially, there is no discharge from the catchment, implying
that
$$\overline{Q}(0) = 0$$   (22a)
$$q(0) = 0$$   (22b)
The solution of Eqs. (6a) and (22a) for the mean runoff discharge is in the form
$$\overline{Q}(t) = \frac{\overline{R}}{K} \int_0^t e^{-(t-y)/K} dy = \overline{R}(1 - e^{-t/K})$$   (23)



It is easy to see from Eq. (23) that the mean discharge decreases with a larger storage
parameter.
We proceed to derive the variance of catchment discharge. A similar procedure to
the above, applying the nonstationary spectral representation for the perturbed quantities
$q(t)$
$$q(t) = \int\limits_{-\infty}^{\infty} \Lambda_{q\xi}(t;\omega)\, e^{i\tau}\, dZ_{\xi}(\omega) \tag{24}$$
and Eq. (11) into Eqs. (6b) and (22b), leads to the following results
$$\frac{d\,\Lambda_{q\xi}}{dt} + (\frac{1}{K} + i\omega)\,\Lambda_{q\xi} = \frac{\Lambda_{r\xi}}{K} \tag{25a}$$
with
$$\Lambda_{q\xi}(0;\omega) = 0 \tag{25b}$$
The solution to this problem is
$$\Lambda_{q\xi}(t;\omega) = \frac{1}{K} \int\limits_{0}^{t} \exp[-\frac{1+i\omega K}{K}(t-y)]\,\Lambda_{r\xi}(y;\omega)dy$$
$$= \frac{1}{2}\frac{e^{-i\tau}}{\alpha(\alpha^2+\omega^2)}\Big[\frac{\alpha-i\omega}{\beta-1}\lambda_1 - \frac{\alpha+i\omega}{\beta+1}\lambda_2 + 2\frac{\alpha}{1+i\omega K}(e^{-\mu}-e^{-i\tau})\Big] \tag{26}$$
where $\lambda_1 = \exp(-\mu)-\exp(-\eta)$, $\lambda_2 = \exp(-\mu)-\exp(\eta)$, $\beta = \alpha K$, and $\mu = t/K$. Eqs. (24) and (26)
provide the framework required to express the discharge perturbation $q(t)$.
The variance of runoff discharge $\sigma_q^2(t)$ can now be obtained as follows:
$$\sigma_q^2(t) = E[q(t)q^*(t)] = \int\limits_{-\infty}^{\infty} \left|\Lambda_{q\xi}(t;\omega)\right|^2 S_{\xi\xi}(\omega)d\omega = \int\limits_{-\infty}^{\infty} S_{qq}(\omega)d\omega \tag{27}$$



where the evolutionary spectrum of $q(t)$ is given by
$$S_{qq}(t;\omega) = \frac{1}{4}\frac{1}{\alpha^2(\alpha^2+\omega^2)^2}\left\{\frac{\alpha^2+\omega^2}{(1-\beta)^2}\lambda_1^2 + 2\frac{\alpha^2-\omega^2}{1-\beta^2}\left[e^{-\mu}(\lambda_1-e^\eta)+1\right] - 4\frac{\alpha}{1-\beta}\left[\frac{\alpha+K\omega^2}{1+K^2\omega^2}\lambda_1(e^{-\mu}-\cos(\tau))\right.\right.$$
$$\left.+\frac{\omega(1-\beta)}{1+K^2\omega^2}\lambda_1\sin(\tau)\right] + \frac{\alpha^2+\omega^2}{(1+\beta)^2}\lambda_2^2 - 4\frac{\alpha}{1+\beta}\left[\frac{\alpha-K\omega^2}{1+K^2\omega^2}\lambda_2(e^{-\mu}-\cos(\tau)) - \frac{\omega(\beta+1)}{1+K^2\omega^2}\lambda_2\sin(\tau)\right]$$
$$\left.+4\frac{\alpha^2}{1+K^2\omega^2}\left[e^{-2\mu}-2\cos(\tau)e^{-\mu}+1\right]\right\}S_{\xi\xi}(\omega) \tag{28}$$
The discharge variance follows from Eq. (27) through the application of Eqs. (18) and

(28):

$$\sigma_q^2(t) = \frac{\pi}{2}I_\xi\frac{1}{\alpha^3}\left[\frac{\lambda_1}{(1-\beta)^2}\left(\frac{\lambda_1}{2}-e^{-\eta}\right) + \frac{\phi_1}{(1+\beta)^2} - \frac{(1+3\beta)\lambda_1 e^{-\mu}}{(1-\beta)(1+\beta)^2} - \frac{e^{-\eta}\phi_2}{1-\beta^2}\right.$$
$$\left. + 4\frac{\beta^2}{(1-\beta^2)^2}(\lambda_1 e^{-\eta} - \beta e^{-2\mu})\right] \tag{29}$$
with
$$\phi_1 = 1 + 2\beta + \frac{1+4\beta}{2}e^{-2\mu} + \frac{e^{2\eta}}{2} - e^{-2\eta} + e^{\eta-\mu} \tag{30a}$$
$$\phi_2 = \eta(\lambda_1+\lambda_2) + \lambda_2 + 2(\eta+1)e^{-\mu} - \eta e^\eta(1-e^{-2\mu}) \tag{30b}$$
Finally, using the relation (19) leads to
$$\sigma_q^2(t) = \frac{\sigma_r^2}{\Gamma_t}\left[\frac{\lambda_1}{(1-\beta)^2}\left(\frac{\lambda_1}{2}-e^{-\eta}\right) + \frac{\phi_1}{(1+\beta)^2} - \frac{(1+3\beta)\lambda_1 e^{-\mu}}{(1-\beta)(1+\beta)^2} - \frac{e^{-\eta}\phi_2}{1-\beta^2}\right.$$
$$\left. + 4\frac{\beta^2}{(1-\beta^2)^2}(\lambda_1 e^{-\eta} - \beta e^{-2\mu})\right] \tag{31}$$
The result of this type can be used directly to evaluate the uncertainty in the mean runoff
discharge model when applying it to the field situations.

Figs. 2a and 2b display the runoff discharge variance in Eq. (31) as functions of the

storage parameter $K$ and rainfall parameter $\alpha$, respectively, for various time scales. It is





seen from Fig. 2a that the discharge variability increases with a decrease in $K$ for a
given $\alpha$. This can be attributed to that persistence of random discharge fluctuations is
reduced by a large $K$, which leads to smaller deviations of the discharge fluctuations. A
similar conclusion has been made for the case of response of the Brownian particle
motion to a stationary random noise forcing. Note that Eq. (6b) is in fact a generalized
Langevin equation (e.g., van Kampen, 1981; Gardiner, 1985) arising in the analysis of
Brownian motion, where $K$ corresponds to a particle mass. It has been reported from the
literature that the velocity variability of the Brownian particle is reduced by a large
particle mass. That is, velocity fluctuations in stationary flow fields persist shorter with a
larger particle mass.
In addition, Fig. 2b shows the reduction in the variability of the runoff discharge field
with $\alpha$ for a fixed value of $K$. It is evident from Eq. (26) that in a linear system, the
variability of output process correlates positively with that of input process. The larger
the rainfall parameter, the smaller the variability of the rainfall field (Fig. 1), and,
consequently, the smaller the variability of runoff discharge (Fig. 2b). In other words, the
runoff processes in response to rainstorms characterized by a small rainfall parameter
exhibit a relatively smoother data profile.

**5  Concluding remarks**

In this work, the catchment-scale rainfall-runoff process is modeled by a linearized model
and analyzed by means of a stochastic framework. In our derivation, the temporal
distribution of the random rainfall process is described by an AR model. The closed-form



solutions to the linear lumped rainfall-runoff model are expressed in terms of first two
statistical moments through the nonstationary Fourier-Stieltjes representation. The first
moment (mean) is used as an unbiased estimate of runoff discharge, while the second
moment (variance) gives a quantitative measure of the uncertainty by applying the mean
rainfall-runoff model to the field situations.
The analysis of the closed-form solutions clearly demonstrates that an introduction of
a large rainfall parameter leads to the reduction in the variability of the rainfall process.
The smaller the storage or rainfall parameters, the more persistence of the random
fluctuations in runoff discharges and, in turn, the larger deviations from the mean, which
results in larger variability of the runoff process.

*Acknowledgements*.      The work underlying this research is supported by the Ministry
of Science Technology under the grants MOST 105-2221-E-009-043-MY2, and
105-2811-E-009 -040.

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

**Figures**

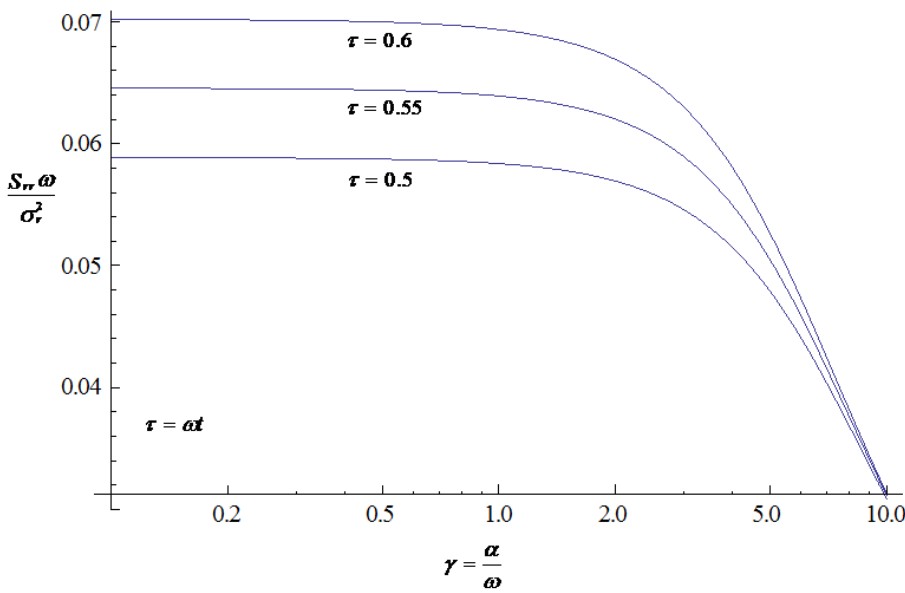


**Figure 1.** The dependence of $S_{rr}(t;\omega)$ in Eq. (20) on rainfall parameter $\alpha$ at different
times.





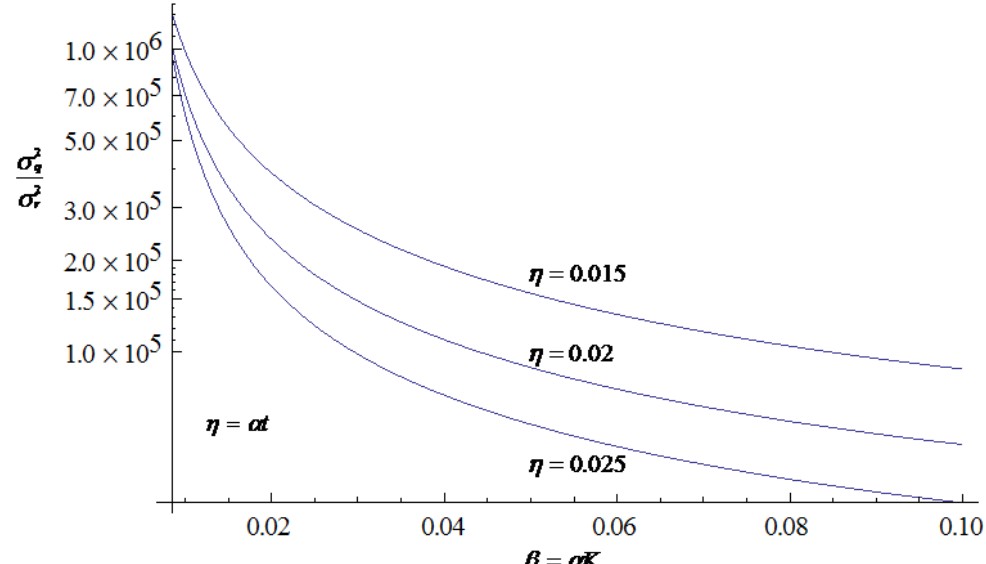



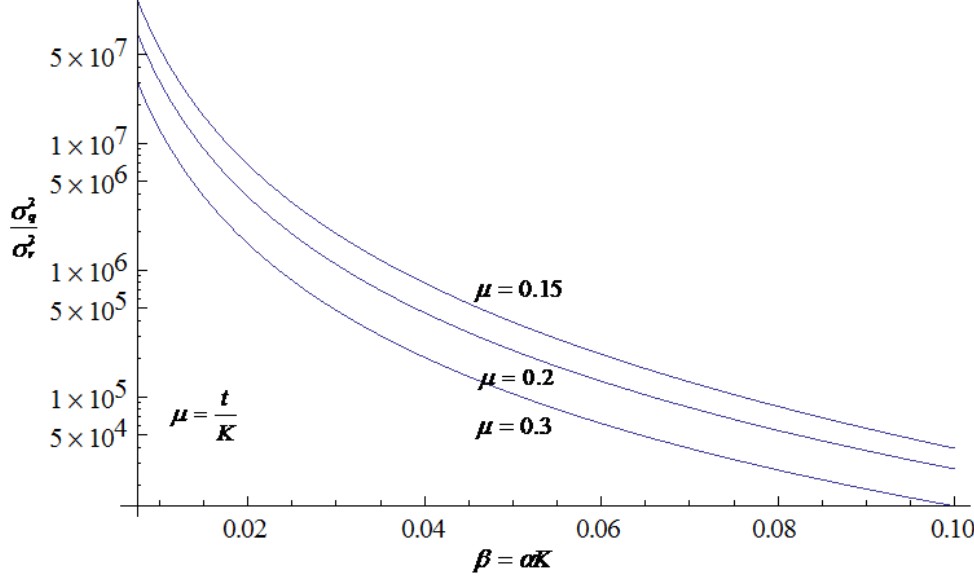


**Figure. 2** The dependence of $\sigma_q^2$ in Eq. (31) on (a) storage parameter $K$ and (b) rainfall
parameter $\alpha$ at different times.