# Peer review of "Published: 23 January 2017"

_Hydrology and Earth System Sciences, 2017_

## Short Comment (SC1) · 28 Jan 2017

Title: Uncertainty quantification in application of linear lumped rainfall-runoff models
Authors: Ching-Min Chang and Hund-Der Yeh Journal: Hydrology and Earth System
Sciences

Review:

To better understand the governing hydrological processes and solve real-world problems in the ambience of constraints, the hydrologic community has been blessed with mathematical models that are formed by translating the processes into mathematical equations subject to boundary conditions. To date, as underscored in the literature,

there are many mathematical models in the field of hydrology. Though there are many classifications to pool these mathematical models, most of the models are pooled under lumped and distributed models. In the absence of extensive data and the thorough understanding of the governing processes, lumped models are favored to get some insight about the problem of interest within a short of period of time and the other binding constraints.

In this manuscript, the authors propose a stochastic framework coupled with a lumped model on rainfall-runoff processes, to analyze the temporal variability of runoff. Based on mathematical derivations, starting from a continuity equation, authors assert that the random variation of the runoff is forced by the temporal variation of rainfall. With this assertion, initially, the authors fit an autoregressive model to account for the temporal variability of the rainfall. Subsequently, the first two moments (i.e., mean and variance) of the runoff are used to analyze the temporal variability of runoff.

Based on this research, the authors conclude that the temporal variability of the runoff induced by a random rainfall process persists longer for smaller values of the storage or rainfall parameters.

Based on this review, the following comments are made:

1) As per the current version of the paper, the sentences in the abstract are scattered. From the reader's point of view, the abstract would be more concrete if the authors streamline the sentences to underscore the research carried out in this paper.

2) In this paper, the evapotranspiration is the only abstraction from the catchment (line number 58 on page number 3). However, further simplification leads to drop the evapotranspiration from the continuity equation (line number 82 on page number four). In other words, the inflow (i.e., rainfall), outflow (i.e., runoff), and the storage are the only components of the system. The authors should clearly state the validity of this conceptualization. Otherwise, the title of the paper may mislead considering the fact that the conceptualized system does not account for all the inner details. Are the authors

formulating the system to derive a solution that is feasible in a mathematical environment?

3) Does the R.H.S of equation 7b need a sign? If the sign is absorbed within the function, the authors need to mention it in the manuscript. The equation 8 also needs to be checked.

4) As per the authors, in most practical applications, S in Eq. (1) is specified as an arbitrary function of Q. To convince this statement, few journal papers need to be cited.

5) As per the current version of the paper, all variables and parameters in equation-1 represent spatial averages over the entire catchment area (line number 62 on page number three). What is meant by variables? What is meant by parameters?

6) As per the authors, to carry out rainfall-runoff calculations detailed information about landscape properties and hydrologic states must be known in the whole catchment (line number 31 on page number two). Authors also state that such information is not available due to the heterogeneity in associated parameters (line number 33 on page number two).What is meant by heterogeneity in associated parameters? What are those associated parameters? Is it due to the heterogeneity that we do not have these information? Recently, it has been argued that the advances in data-intensive hydrologic science have laid the foundation for a data-driven hypothesis testing framework (http://www.hydrol-earth-syst-sci-discuss.net/hess-2016-695/). Therefore, the authors need to convince their statement.

7) As per the authors, referring to line number 66 on page number 3, there are two unknowns, namely Q and S. What has motivated the authors to assert a statement of this nature? Is Et known? If Et is known, what has driven the authors to consider Q as unknown.

8) From the reader's point of view, the title of section 2 is meaningless. What is meant by "problem"?
9) As per the current version of the paper, "rainstorm" is the major input into the generation of surface runoff and the production of runoff (line number 28 on page number two).From the reader's point of view, this statement is not warranted. What is meant by "major input"?

10) As per the current version of the paper, the authors propose a stochastic framework for a linear lumped rainfall-runoff problem at the catchment scale (line number nine on page number one). The authors should clearly state this rainfall-runoff problem.

11) As per the current version of the paper, the title of the paper is uncertainty quantification in application of linear lumped rainfall-runoff models. However, the abstract of the paper does not explicitly present about the uncertainty quantification.

12) How is the outcome of this research influenced if a non-linear relationship between the storage and the outflow is assumed? It would be more useful if the authors discuss on this.

http://research.abzwater.com/review/ABZR5.pdf

---

## Referee Comment (RC1) · G. Pegram (Referee) · 3 Feb 2017

The paper contains virtually no hydrology but is stuffed with 'mathematicity', some of it over complicated. The 2 figures (there are only 2) display the link between theoretical parameters derived from a linear model, but there is not one figure showing rainfall and the attendant runoff. There is no mention of Dooge, Nash nor Diskin. I would NOT recommend it for HESS.

Geoff Pegram

---

## Author Comment (AC1) · 17 Feb 2017

Please see the attached files.

Please also note the supplement to this comment:
http://www.hydrol-earth-syst-sci-discuss.net/hess-2017-19/hess-2017-19-AC1-supplement.zip

---

## Author Comment (AC2) · 24 Feb 2017

Please see the attached files.

Please also note the supplement to this comment:
http://www.hydrol-earth-syst-sci-discuss.net/hess-2017-19/hess-2017-19-AC2-supplement.zip

---

## Short Comment (SC2) · 6 Mar 2017

This technical note develops analytical expressions for the first two moments of rainfall and runoff time-series. I.e. to model a time series of the mean and variance of rainfall and runoff. The main assumptions, which are clearly noted, are that evaporation is negligible, the rainfall is an autoregressive process, the rainfall-runoff is a linear storage, and the initial rainfall and runoff are zero.

The technical note is generally well written: concise and clear. I lack expertise to comment on the accuracy of the mathematical solutions, however assuming they are accurate, they are clearly and concisely presented. However, I can query the derivation of Eq. 8 from 7b: the right hand side of 7b also needs divided by 'a'? Therefore, the

authors should re-inspect the accuracy of this and all equations.

The practical value is limited by the assumptions. In particular the method is only applicable to storm events in catchments where the linear rainfall-runoff relationship suffices. The realism of the AR rainfall model is questionable. No validation is attempted, in fact no time-series results are shown, so it's difficult for the reader to judge whether or not the results are plausible. No critical discussion of applicability is provided.

I note the comment of the first reviewer that there is no mention of some of the classical work on linear rainfall-runoff modelling, and certainly the technical note should be put in context of these established methods.

I am also not expert enough to comment on the novelty of applying the nonstationary Fourier-Stieltjes representation to rainfall-runoff. If indeed this is novel, it seems to be a neat analytical solution, worthy of publication as a technical note.

Therefore assuming the equations are correct and this is a novel approach to rainfall-runoff analysis, I recommend major revisions, consisting of adding context, an illustration of time-series results, a brief critical discussion of applicability.

Neil McIntyre

---

## Author Comment (AC3) · 20 Mar 2017

Please see the attached files.

Please also note the supplement to this comment:
http://www.hydrol-earth-syst-sci-discuss.net/hess-2017-19/hess-2017-19-AC3-supplement.zip

---

## Author Comment (AC4) · 20 Mar 2017

Please see the attached files.

Please also note the supplement to this comment:
http://www.hydrol-earth-syst-sci-discuss.net/hess-2017-19/hess-2017-19-AC4-supplement.zip